# Micro-/Nano-Structured Biodegradable Pressure Sensors for Biomedical Applications

**DOI:** 10.3390/bios12110952

**Published:** 2022-11-01

**Authors:** Yoo-Kyum Shin, Yujin Shin, Jung Woo Lee, Min-Ho Seo

**Affiliations:** 1Department of Information Convergence Engineering, Pusan National University, 49 Busandaehak-ro, Mulgeum-eup, Yangsan-si 50612, Gyeongsangnam-do, Korea; 2Department of Materials Science and Engineering, Pusan National University, 2 Busandaehak-ro 63beon-gil, Geumjeong-gu, Busan 46241, Korea; 3School of Biomedical Convergence Engineering, Pusan National University, 49 Busandaehak-ro, Mulgeum-eup, Yangsan-si 50612, Gyeongsangnam-do, Korea

**Keywords:** biodegradable electronics, pressure sensors, MEMS (microelectromechanical systems), NEMS (nanoelectromechanical systems), biomedical engineering

## Abstract

The interest in biodegradable pressure sensors in the biomedical field is growing because of their temporary existence in wearable and implantable applications without any biocompatibility issues. In contrast to the limited sensing performance and biocompatibility of initially developed biodegradable pressure sensors, device performances and functionalities have drastically improved owing to the recent developments in micro-/nano-technologies including device structures and materials. Thus, there is greater possibility of their use in diagnosis and healthcare applications. This review article summarizes the recent advances in micro-/nano-structured biodegradable pressure sensor devices. In particular, we focus on the considerable improvement in performance and functionality at the device-level that has been achieved by adapting the geometrical design parameters in the micro- and nano-meter range. First, the material choices and sensing mechanisms available for fabricating micro-/nano-structured biodegradable pressure sensor devices are discussed. Then, this is followed by a historical development in the biodegradable pressure sensors. In particular, we highlight not only the fabrication methods and performances of the sensor device, but also their biocompatibility. Finally, we intoduce the recent examples of the micro/nano-structured biodegradable pressure sensor for biomedical applications.

## 1. Introduction

A pressure sensor is a device that can transduce mechanical pressure to an electrical or optical signal. It has gained considerable attention from various traditional industries such as automotive [1,2], aviation [3,4,5], and manufacturing industries [6,7,8,9] since it can help in controlling the system stability and process flow. Recently, owing to the advances in materials science, mechanical engineering, and fabrication technologies, the pressure sensor can be implemented with various materials and dimension, enabling its application in a wide range of emerging academia disciplines and industrial fields, such as robotics [10,11,12,13,14,15,16,17], artificial intelligence [18,19,20,21], and smart factory [22,23]. Among them, the use of pressure sensors in the biomedical engineering field is one of the representative promising applications of this technology because it can help to measure the pressure of the target tissue and organ that is essential for wide range of disease diagnosis and therapy (Figure 1a,b) [24,25,26,27,28,29].

Conventionally, bio-implantable pressure sensors have been made of materials being able to permanently exist in human-body with bio-compatibility. However, these types of devices require secondary surgical extraction after clinical use, and it intrinsically accompanies the risk, cost, distress, and pain for the patient [30]. In this regard, biodegradable (or, equivalently, bioabsorbable and bioresorbable) pressure sensor has recently received lots of attention from researchers because it can be degraded and disappear in human body spontaneously by bio-fluids after specific period [31,32,33,34].

Earlier developed biodegradable pressure sensors were partially composed of biodegradable materials with a simple structure [35,36]. However, owing to the recent developments in materials science, mechanical and electrical engineering and fabrication technologies, biodegradable pressure sensors have recently become fully biodegradable with high scalability [37,38]. In particular, as the device is designed to exploit micro-/nano-structures, it can not only be manufactured in a miniature dimension making it easier to implant, but it also shows great improvement in performance when used for precise detection of bio-signals and diseases [30,39,40]. As a result, fully biodegradable and miniaturized pressure sensors with improved performance in terms of high sensitivity, fast response time, stability, and reliability have been realized, and this has made biodegradable pressure sensors usable in diagnosis and healthcare applications.

Even though there has been a sudden spurt in progress on these sensors, however, there has been less reports to summarize and review the recent progress and knowledge regarding the micro-/nano-structured biodegradable pressure sensors and their applications. Here, this review paper introduces the latest progress in the development of micro-/nano-structured biodegradable pressure sensors. We first introduce recent developments and libraries in biodegradable materials for the micro-/nano-structured biodegradable pressure sensors; in particular, materials are classified as wet and dry transient materials, depending on the degradation conditions. Next, we introduce details about the performance-enhancement in the biodegradable pressure sensor exploiting the micro-/nano-structure. Especially, we focus on the characteristics, performance, and fabrication methods of these sensors. Finally, the practical applications of the micro-/nano-structured biodegradable pressure sensor in biomedical engineering are introduced. We review the reports on the feasibility of the developed micro-/nano-structured biodegradable pressure sensors for healthcare and diagnosis of disease in wearable and implantable biomedical applications without any biocompatibility issue.

## 2. Transient Materials

To develop biodegradable electronics with a transient performance (Figure 2), appropriate selections of materials should be considered from the viewpoint of electronic components (e.g., conductors, semiconductors, insulators, encapsulations, and substrates). These materials are required to satisfy designated dissolution rates, electrical/mechanical properties, and other demands depending on the desired application of the device. In the following sections, we summarize the various types of inorganic and organic biodegradable materials that are commonly utilized in recent research on transient electronic devices. The materials are classified into two parts namely wet and dry transient materials, according to their degrading mechanisms.

### 2.1. Wet Transient Materials

#### 2.1.1. Conductors

Alkaline earth metals including magnesium (Mg) and calcium (Ca) along with transition metals such as molybdenum (Mo), tungsten (W), iron (Fe), and Zinc (Zn) are considered as potential conductors for transient devices owing to their high hydrolysis/dissolution rates, easy fabrication process, and their biocompatibility along with their intrinsic electrical conductivity [32,43,44,45,46,47]. Mg and Fe are the most promising candidates that could be used as in vivo conductors in electronic devices because of their high biocompatibility [48,49]. These inorganic materials are essential to keep the human body healthy and functioning properly. According to the National Institute of Health (NIH), the recommended dietary allowance (RDA) values of Mg and Fe are 310–420 mg and 8–18 mg each for adults older than 19 [50,51]. In other words, when these metals are taken in or absorbed in the right amount they have no side effects on the human body. Thus, Mg, Mg alloy, and Fe are widely used as materials for bioresorbable implants such as vascular stents [44,48]. In general, the aforementioned biodegradable metals can dissolve in an aqueous solution and produce metal cations. Then, the metal cations may form metal hydroxides or metal oxides via hydrolysis as described in the following reaction procedures [43,52,53,54,55,56]:$$\mathrm{Mg}+2H_{2}O\to \mathrm{Mg}{\left(\mathrm{OH}\right)}_{2}+H_{2}$$
$$\mathrm{Fe}+2H_{2}O\to \mathrm{Fe}{\left(\mathrm{OH}\right)}_{2}+H_{2}$$
$$\mathrm{Zn}+2H_{2}O\to \mathrm{Zn}{\left(\mathrm{OH}\right)}_{2}+H_{2}$$
$$\;2W+2H_{2}O+3O_{2}\to 2H_{2}{\mathrm{WO}}_{4}$$
$$2\mathrm{Mo}+2H_{2}O+3O_{2}\to 2H_{2}{\mathrm{MoO}}_{4}$$

Figure 3a illustrates the degrading behavior of the aforementioned metals. The dissolution rates in simulated body fluids of Mg, Fe, Zn, W are ≈0.05–0.5, ≈0.02, ≈0.005, ≈0.02–0.06 μm/h, respectively [43]. Each material shows different dissolution kinetics and distinctive advantages. In certain cases, specific electronics must disintegrate as soon as they start to react, whereas others are required to maintain their shapes and functions until the targeted time. Hence, appropriate materials should be selected for the transient devices depending on the aim of the reaction and the operating time. For instance, metals based on Mg have relatively high dissolution rate; hence, they would require encapsulation to reduce the rate of disintegration [57]. Moreover, methods to control the dissolution rate of Mg have been studied such as alloying with aluminum (Al) (e.g., AZ31B Mg alloy, Mg-Al-Zn alloy) or coating with biocompatible ceramics such as Ca-P or MgF_2_ [58,59,60]. For example, the electrical dissolution rate of AZ31B Mg alloy is three times lower than Mg. By contrast, transition metals such as Mo have relatively low dissolution rate; hence, they can delay disintegration upto the targeted time and function normally in vivo without any encapsulation [43,52].

#### 2.1.2. Semiconductors

Semiconducting materials are essential in most electronic devices, such as transistors, sensors, diodes, power harvesting devices [63,64,65]. However, traditional semiconducting materials when used for transient electronics are limited by their toxicity and rigidity [66]. For instance, gallium arsenide (GaAs) is a traditional compound semiconducting material that is widely utilized in microcircuits owing to its distinct advantage in increased electron mobility. However, several studies have demonstrated the acute and chronic toxicity of GaAs and the damage caused by GaAS to the lungs, reproductive organs, and kidneys of animals after inhalation exposure [67,68,69]. Therefore, transient semiconducting materials are still challenging and are many studies have been conducted to overcome these drawbacks.

##### Inorganic Materials

Among the various semiconducting materials, silicon (Si) is one of the most common. Bulk Si is usually considered to be non-degradable; traditional devices have been fabricated as thick wafers so that the integrated circuits do not decompose for several hundred years [53,70]. However, when its structure is scaled down to the nanoscale, it could be fully biodegraded in aqueous solutions or biofluids (Figure 3b). Furthermore, the compatibility of Si with widely used fabricating processes such as photolithography is an advantage in utilizing it for transient devices. Hwang et al. reported the dissolution property of a single-crystalline Si nanomembrane (Si NM), on a silicon-on-insulator wafer with a lateral dimension of 3 μm × 3 μm and a thickness of 70 nm, in phosphate-buffered saline (PBS; pH of 7.4). The dissolution rates at room temperature (25 °C) and body temperature (37 °C) were 2 nm/day and 4.5 nm/day, respectively [32]. These rates could vary in the range of 0.5–624 nm/day depending on the crystallinity, morphology and doping level of Si or on the surroundings such as the ambient temperature or composition of the solution [71]. In general, Si undergoes hydrolysis to produce orthosilicic acid (Si(OH)_4_) as described in the following reaction (Figure 3c) [32,72].
$$\mathrm{Si}+4H_{2}O\to \mathrm{Si}{\left(\mathrm{OH}\right)}_{4}+2H_{2}$$

Besides Si NM, dissolution behavior and biocompatibility of other semiconducting materials such as amorphous silicon (a-Si), polycrystalline silicon (poly-Si), alloys of silicon and germanium (SiGe), and germanium (Ge) have also been investigated. By measuring the thickness of a patterned array of squares (3 μm × 3 μm × 100 nm) of each material, the dissolution rate of a-Si, poly-Si, SiGe, and Ge in buffer solutions (pH 7.4, 37 °C) are investigated as 4.1, 2.8, 0.1, and 3.1 nm/day, respectively. Compared to poly-Si, the dissolution rate of a-Si is accelerated owing to its low-density structure, which facilitates diffusion of aqueous solutions [52].

In addition, semiconducting oxides including zinc oxide (ZnO) are alternatives to Si owing to their high carrier mobility and outstanding transparency in the visible wavelength range. Furthermore, as ZnO shows degrading behavior in aqueous conditions, it is used as a semiconducting material in transient electronics. ZnO involves hydrolysis to form zinc hydroxide (Zn(OH)_2_). For example, a 200 nm-thick ZnO completely disappeared in DI water at room temperature within 15 h (Figure 3d) [41]. Furthermore, ZnO can be dissolved in ammonia (pH of ~7.0 and ~9.0) and NaOH solutions (pH of 7.0 and ~9.0). Following are the chemical reactions of ZnO during dissolution in DI water, ammonia, and NaOH solution [73]:$$\mathrm{ZnO}+H_{2}O↔\mathrm{Zn}{\left(\mathrm{OH}\right)}_{2}$$
$$\mathrm{ZnO}\left(s\right)+2H^{+}\to {\mathrm{Zn}}^{2+}+H_{2}O$$
$$\mathrm{ZnO}\left(s\right)+4{\mathrm{HN}}_{3}\cdot H_{2}O\to \mathrm{Zn}{\left({\mathrm{NH}}_{3}\right)}_{4}^{2+}+2{\mathrm{OH}}^{-}+3H_{2}O$$
$$\mathrm{ZnO}\left(s\right)+2{\mathrm{OH}}^{-}\to {\mathrm{ZnO}}_{2}^{2-}+H_{2}O$$

Zhou et al. reported that ZnO was visibly etched in about one hour in horse blood serum. This indicates that if a ZnO wire is trapped in a blood vessel or in the body, it would dissolve into Zn ions that could be absorbed by the surroundings and become a part of the nutrition of the body. Notably, Zn ions are needed every day for the human body to function properly [73].

##### Organic Materials

Polymer materials are also widely used as semiconducting material for transient devices because of their soft texture, flexible mechanical properties, low cost, and their suitability for large-scale production. These polymers could be categorized into synthetic and natural based polymers.

First, several synthetic biodegradable polymers such as poly(diketopyrrolopyrrole-*p*-phenylenediamine (PDPP-PD) and 5,5′-bis-(7-dodecyl-9*H*-fluoren-2-yl)-2,2′-bithiophene (DDFTTF) are currently utilized in the fabrication of soft electronics [74]. Diketopyrrolopyrrole(DPP) is a precursor of many synthetic polymers. Its monomer originated from a natural resource and hence, would also be degradable [49,75,76]. The PDPP-PD is a semiconducting polymer synthesized by a condensation reaction between diketopyrrolopyrrole-aldehyde (DPP-CHO) and *p*-phenylenediamine, under catalysis, using *p*-toluenesulfonic acid (PTSA). The chemical structure formula of PDPP-PD is illustrated in Figure 3e. PDPP-PD is fully disintegrable and biocompatible based on reversible imine chemistry. Under neutral-pH conditions, the imine bond (–C=N–) preserves a stable conjugated linker; however, it can be easily hydrolyzed by adding a trace amount of acid. A device consisting of this conjugated polymer degraded completely within 30 days in a pH 4.6 buffer solution [49]. DDFTTF is a robust small-molecule *p*-channel semiconducting material that indicates outstanding device performance. Bettinger et al. reported that a DDFTTF layer that was exposed in a citrate buffer delaminated in less than two days; thus, the device lost its functionality, irreversibly [34].

Natural polymers such as indigo and melanin are also utilized owing to their excellent biocompatibility. Indigo is an intrinsically ambipolar organic semiconducting material with a bandgap of 1.7 eV and high electron and hole mobilities of 1 × 10^−2^ cm^2^/V·s [77]. Because of the natural existence of indigoids (derivatives of indigo), which have good semiconducting performance, low toxicity, and chemical stability, indigo is one of the most promising candidates for biocompatible semiconducting materials [78]. Eumelanins are a subclass of melanins. They exhibit excellent biocompatibility along with biodegradability via free radical degradation mechanisms. Furthermore, eumelanin can serve as a biocompatible electrode in high-density charge storage devices due to its unique properties. For instance, they show a hybrid ionic-electronic conduction behavior by the self-doping mechanism which is hydration-dependent [79,80].

#### 2.1.3. Insulators, Encapsulations, and Substrates

Other important transient materials for electronic components are insulating materials that can isolate conductors and semiconductors [32,41]. In addition, substrate materials for most biodegradable electronics or devices, in thin geometry, should be suitable for handling and integrating all necessary components. The lifespan of transient electronic devices depends highly on the dissolution behavior of bioresorbable materials, which make up the device. Therefore, in some cases, encapsulation layers are necessary to enable stable and programmable operation of the devices by protecting the active layers from direct contact with the ambient aqueous environments, thus preserving the electrical functions of the devices [44,55,81]. Several inorganic/polymer materials have been widely adopted for this purpose. In general, oxides and nitrides of Si, Mg, and some polymers are used as insulators, encapsulations, or substrates for transient electronics owing to their bioresorbability and compatibility with the vacuum deposition processes and photolithography [39,53,82].

Yu et al. reported a fully bioresorbable, thin, flexible neural silicon electronic array, which could record electrophysiological signals in vivo. This embodiment uses SiO_2_ for gate dielectrics, a tri-layer of SiO_2_/Si_3_N_4_/SiO_2_ for encapsulation, and a 30 μm-thick flexible sheet of poly(lactic-*co*-glycolic acid) (PLGA), a bioresorbable polymer, as the substrate. SiO_2_ and Si_3_N_4_ dissolve at a rate of ~8.2 nm/day and ~5.1 nm/day, respectively, in biofluids at pH 7.4 and temperature 37 °C; PLGA completely dissolves within 4 or 5 weeks in biofluids at 37 °C [81]. SiO_2_, SiN_x_, and MgO are consumed by hydrolysis according to following reaction procedures [32,83]:$${\mathrm{SiO}}_{2}+2H_{2}O\to \mathrm{Si}{\left(\mathrm{OH}\right)}_{4}$$
$${\mathrm{Si}}_{3}N_{4}+6H_{2}O\to 3{\mathrm{SiO}}_{2}+4{\mathrm{NH}}_{3}$$
$$\;\mathrm{MgO}+H_{2}O\to \mathrm{Mg}{\left(\mathrm{OH}\right)}_{2}$$

MgO, specially, is widely used as an encapsulating oxide layer in implantable devices such as endovascular bioresorbable stents, to control the rate of degradation of the active agents, owing to its adequate biocompatibility [44]. The dissolution kinetics of MgO was investigated concerning the Mg^2+^ and H^+^ concentration. For example, a solution where pH is larger than 7 at room temperature leads to a maximum dissolution rate [84]. Moreover, thermoplastic polyesters such as PLGA and water-soluble polymers including poly(vinyl alcohol) (PVA) are commonly used biodegradable polymers for medical implants and drug delivery systems [39,85,86]. As PGLA and PVA have the advantages of price, easy access for use, and easy fabrication, they have been used in a variety of devices for biomedical and biotechnology applications such as surface-mediated drug delivery and tissue engineering [87,88,89]. For instance, PLGA is used as a substrate for organic thin-film transistors, whereas PVA is used as the gate dielectric of the transistors [34]. In addition, natural silk fiber is also considered as a promising material for substrates or encapsulations owing to their outstanding mechanical robustness, biocompatibility, flexibility, ease of processing, and programmable biodegradability from minutes to years [49,66,90]. For example, Tao et al. reported a fully dissolvable wireless heating device operating in vivo to provide the necessary thermal therapy of trigger drug delivery. Here, an Mg resistor, an MgO dielectric layer, and an Mg coil are deposited onto a silk substrate. Further, all the layers were encapsulated in a silk overcoat, which could be used to adjust the lifespan of the device [91].

Besides the aforementioned biomedical applications, energy harvesting devices such as batteries that utilize the transient behavior of some materials have been studied (see Figure 3f) [62,92].The term “transient” here could be considered to mean that even though certain parts of the materials continue to exist, individual parts of the device could break apart and these devices do not function properly any longer. For example, Fu et al. worked on a transient battery that consisted of vanadium oxide (V_2_O_5_) as cathode, polyvinylpyrrolidone (PVP) as separator, sodium alginate (Na-AG) as battery encasement, and aluminum (Al), copper (Cu) as current collector [62].

### 2.2. Dry Transient Materials

In addition, materials that can spontaneously degrade in dry conditions or whose deterioration can be be triggered by a certain stimulus have been studied. They are called ‘dry transient materials’, whereas wet transient materials are materials that degrade when submerged in an aqueous solution or biofluid. Therefore, dry transient materials have the advantage of being suitable for use in a broad range of operating environments without the requirement of microfluidics. In general, they are utilized in encapsulations and substrates for integrating active electronics within transient devices. Beyond these characteristics, in the case of stimuli-triggered transient materials, they can provide further benefits, such as the control of the dissolution rate and the lifespan of the system, by triggering condition. Conversely, the dissolution rate of a wet transient device could be only determined by the materials selected and the fabrication procedure.

Cyclododecane (CDD) is a dry-degradable material, which disappears completely through sublimation when continuously exposed to air. Owing to its sublimating nature, it can be applied as a transient layer to protect fragile or sensitive surfaces. As CDD has a high vapor pressure of 0.1 hPa at 20 °C so it could sublimate adequately at room temperature, and has a low melting point (60.8 °C) as shown in Figure 4a. The sublimation rate of CDD could be affected by several conditions. For example, the rate is about 1.3 μm/h in the absence of ventilation, whereas it increases to 4.6 μm/h under ventilation (under air velocity at 1.9 m/s). In addition, it could be effectively tuned by adding 3–15% *w*/*w* of titanium dioxide nanoparticles (TiO_2_ NPs, 10 nm of average diameter). This leads to an increase in sublimation enthalpy owing to the reduction of mesoscale extension of volatile molecular layers, resulting in reduced rate of mass loss. For example, the CDD samples doped with 8% TiO_2_ NPs are approximately twice as thick as those of pure CDD after 500 h of sublimation [42].

In addition, poly(phthalaldehyde) (PPA) is an acid-sensitive metastable polymer, which has a low ceiling temperature (T_c_ = −43 °C); therefore, it can be easily synthesized with various end-groups and it rapidly depolymerizes upon backbone bond cleavage [93,94]. Hernandez et al. demonstrated photo-triggerable or photo-degradable transient electronics fabricated on a cyclic PPA (cPPA) substrate with a photo-acid generator (PAG) as additive (Figure 4b,c). The electronics were disintegrated by stimulating the PAG/cPPA substrates with a UV source. The rates of transience were controlled by regulating the PAG concentration and the irradiance of UV light. To include UV light sensitivity, 2-(4-methoxystyryl)-4,6-bis(trichloromethyl)-1,3,5-triazine (MBTT) were utilized for PAG additives. When exposed to a UV light with maximum wavelength of 379 nm, the MBTT formed a highly reactive Cl· radical that extracts a hydrogen atom from ambient environment to form hydrochloric acid (HCl). Figure 4c illustrates the photoinduced disintegration mechanism of MBTT/cPPA substrate along with the electronics [93].

Park et al. reported the development of thermal-triggered transient electronics consisting of Mg electrodes and wax coatings, which contain triggerable-encapsulated acid microdroplets on acid-sensitive cPPA as substrate (Figure 4d,e). When exposed to sufficient heat (~55 °C), the device broke down rapidly as the protective wax coatings melted and released the encapsuled acid microdroplets. Rapid destruction due to the acidic depolymerization of cPPA can be induced using cPPA substrates. The overall mechanism of the Mg electrode degradation and cPPA substrate depolymerization are shown in Figure 4d. Furthermore, the destruction time can be controlled by tuning the thickness of the wax protection layer, acid concentration, and trigger temperature [94].

Depending on the materials used and the structure of devices, the range of the degrading temperature is controllable. Si NM electronics integrated with sufficiently thin and high-temperature degradable poly-α-methylstyrene (PAMS) was also suggested by Li et al. (Figure 4f–h). The PAMS layer releases volatile monomers (α-methylstyrene) when it is heated up to ~250–300 °C. The degradation mechanism of PAMS is illustrated in Figure 4h. Assuming that the PAMS layer would decompose completely to its monomers, the maximum pressure in a confined space could be estimated using the ideal gas law as follows:$$P=\frac{nRT}{V}=\frac{\left(\frac{m}{M}\right)PT}{V}=\frac{\rho RT}{M}$$
where *n*, *m*, and *M* is the mole, mass, and molecular weight of the monomer, respectively. Here, density ρ = 1.075 g/cm^3^, gas constant *R* = 8.31 Pa·m^3^/mol·K, temperature *T* = 573 K, and *M* = 118.18 g/mol. In a confined space, the maximum gas pressure could be *P* = 43.34 MPa. The pressure from these gas products causes a high stress (~23 kPa) on the Si NM layer, which suffers a displacement of up to 11.9 μm (see Figure 4g). This distortion in the layers, results in a functional degradation of the device [95].

Owing to the advantages of each of the aforementioned materials, devices that integrate both dry and wet transient materials have been studied. For example, Camposeo et al. presented a dry-wet transient device in photonics, based on fully organic components. It consists of a water-soluble compound layer such as PVA on CDD as the sublimating substrate [96]. In conclusion, by utilizing the dry transient materials for some components of the device which operate outside of the body with bioresorbable materials for the in vivo part, a fully degradable sensor for healthcare monitoring can be developed.

## 3. Mechanism, Development, and Applications of Biodegradable Pressure Sensors

### 3.1. Mechanisms

Biodegradable pressure sensors can be classified, on the basis of their working principle, into piezoresistive, piezocapacitive, piezoelectric and piezo-optical sensors. A piezoresistive pressure sensor is a sensor that indicates a change in pressure applied to the resistor by a change in the resistance as an electrical signal (Figure 5a) [97,98,99]. The mechanism of the piezoresistive type is associated with two main factors. One mechanism is the resistance change by the deformation of a resistor. When an external physical force is applied to the resistor, the dimension of the resistor changes according to Poisson’s ratio and is determined by the material of the resistor [100]. The other mechanism utilizes the changing resistivity of the resistor material, such as the conductive particle–polymer composite. When a force is applied to the conductive particle–polymer composite, the effective resistance value is lowered as the inter-particle spacing between the conductive particles is reduced and the contact area is increased to improve the electrical path (i.e., percolation effect) [101,102,103]. Typically, piezoresistive pressure sensors have the following advantages: low cost, simple structure and working principle, and a relatively easy fabrication process. They are one of the most widely used pressure sensors [104]. Based on the aforementioned advantages, piezoresistive pressure sensors have been widely used in many applications such as automobiles, medical and home appliances, industrial and aerospace instruments [105,106,107,108]. In spite of its many advantages, the piezoresistive type pressure sensor has the disadvantages of relatively slow response time, low durability, cross-talk between temperature and strain as the resistance changes, and higher power consumption than other types of sensors [109]. To develop a high-performance pressure sensor, piezoresistive pressure sensors are being actively investigated, through the use of micro-/nano-structures and material designs, to maximize the deformation and the porosity of the resistor.

The piezocapacitive pressure sensor measures the change in the capacitance (*C*) as electrical signal when pressure is applied to the device (Figure 5b) [110]. The capacitance is dependent on the dielectric constant (*ε*), the distance between the conductor plates (*d*), and the area of the conductor plates (*A*).
$$C=\varepsilon \frac{A}{d}$$

Generally, when a force is applied perpendicularly to the conducting plates of the capacitor, the distance between the plates decreases, the accumulated electric charge between the conductor plates increases and the capacitance value changes. The capacitive pressure sensors present a non-linear response as the sensitivity drops towards high pressure, owing to decreased compressibility at high pressure [111]. The piezocapacitive pressure sensors have the advantage of low hysteresis, good repeatability, fast response times, and insensitivity to temperature changes. In general, capacitive pressure sensors have been widely used in traditional industries such as automobiles and robots. Recent advances in microstructures, materials, and process design have made it possible to develop high performance piezocapacitive sensors that are small, light, and flexible. Therefore, recent capacitive pressure sensors can also be used in the healthcare and wearable devices [112,113,114,115,116,117,118,119].

Figure 5c is a schematic of the mechanism of working of the piezoelectric pressure sensor [110]. Piezoelectric effect is produced by a change in the arrangement of ions in the noncentrosymmetric structure of piezoelectric materials or the molecular structure of a polymer and its orientation (defined as “re-arragement of the material charges”) when the dynamic pressure is applied. The electric charge can be measured as a voltage proportional to the applied pressure. Piezoelectric pressure sensors have the advantage of high frequency and fast response time which are very important in the automotive industry and aerospace fields [120]. However, it is difficult to measure static pressure because the output signal generated from the piezoelectric materials has a characteristic that the voltage gradually drops to zero in the case of constant pressure. Traditionally, piezoelectric pressure sensors have been developed using inorganic materials with high piezoelectric performance such as lead zirconate titanate (PZT). However, PZT, it has the property of toxicity and hence is not considered to be biocompatible. Although organic piezoelectric materials have recently attracted considerable attention use in biocompatible pressure sensors, this material often does not have a comparable piezoelectric output compared to inorganic piezoelectric materials due to intrinsically insufficient, normal or shear, piezoelectric effects. To improve the piezoelectric output of the organic pressure sensor, several research groups have attempted to design micro and nanostructures. For example, Curry et al. reported an organic nano-fiber based piezoelectric pressure sensor. The developed sensor exploits the aligned nanofiber of poly(L-lactic acid) (PLLA) that can maximize the piezoelectric performance with high flexibility based on the high alignment parallel to the input stress [121,122,123].

One of the other representative biodegradable pressure sensors for biomedical application is an optical type that measures the change of intensity or peak wavelength of the input light when pressure is applied (Figure 5d) [24]. Owing to its immunity to electromagnetic interference, the optical pressure sensor can be used to monitor pressure during an MRI scan or RF ablation procedure. The sensor also has other advantages such as the absence of potentially hazardous voltage, low weight, small size, high sensitivity, and large bandwidth. In addition, light intensity based sensors have the characteristic that they are very insensitive to temperature changes because the measurement and reference detectors are equally affected by the temperature and have very low hysteresis and repeatability errors. In particular, since the output signal of the optical type pressure sensor can be made sensitive to micro-/nano-scale changes, the device can be easily miniaturized and used in several applications that can adapt the micro-/nano-structures for the higher performances. In this regard, the miniaturized pressure sensor of the piezo-optical type is suitable for the medical field. The Fabry–Perot interferometer (FPI), using the interference of light reflected between two parallel glass plates, and the Fiber Bragg grating (FBG) using the principle that light propagating in an optical fiber reflects part of the light in a periodic grating, are widely used in medical applications to measure pressure in the blood vessels, lung pressure, bladder, brain, bone, and joint pressure [24,124,125,126].

### 3.2. Development of Various Biodegradable Pressure Sensor Devices

Based on the aforementioned pressure sensing mechanism, biodegradable pressure sensor technologies utilizing biodegradable (or bioresorbable, equivalently) materials have recently attracted considerable attention in the biomedical field, especially in bio-implantable applications. To adapt the biodegradable pressure sensor to biomedical applications, it is fundamentally important to have high sensitivity and fast response times to accurately measure pressure in the targeted bio-pressure range. In addition, a high flexibility for measuring the pressure on a curvilinear body, and power efficiency, stability, and reliability characteristics to reduce power consumption are also important. Although there are various studies to satisfy the above-mentioned requirements, studies using micro-/nano-structures are receiving a lot of attention especially because of their superiority in terms of their miniaturization and reproducibility. In this section, we will introduce the latest technologies for various high-performance biodegradable pressure sensors using micro-/nano-structures technology.

#### 3.2.1. Biodegradable Piezoresistive Pressure Sensor

The first biodegradable piezoresistive pressure sensor was a piezoresistive porous foam using a multi-walled carbon nanotube (MWCNT)-Poly (glycerol sebacate) (PGS) (Figure 6a) [127]. The developed sensor was basically a composite of PGS, a biodegradable material, and MWCNT, which is conductive, as pressure sensing material. In this sensor, the MWCNT-PGS composite deforms according to the pressure, and inter-particle spacing decreases resulting in a change in resistance (percolation effect). Since the developed composite material has a microporous structure inside, it can easily cause mechanical deformation with respect to the applied pressure, thus high sensitivity can be expected (Figure 6b) [127]. One advantage of this method is that the sensitivity can be a design factor. The micro-porous density can be adjusted by the hydrolytic degradation time of the MWCNT-PGS composite, and the sensitivity can be adjusted from 0.12 ± 0.03 kPa^−1^ to 8.00 ± 0.20 kPa^−1^ (Figure 6c) [127].

The material-based piezoresistive pressure sensor has advantages such as a simple fabrication process and structure. However, in this method, the deformation of the material containing the conductive fillers easily reaches the saturation state in the range of higher pressures; hence, the resistance change according to the pressure is very non-linear, and there are limitations in performance such as operation range. To improve the performance of the sensor, researchers have recently conducted a study on designing biodegradable materials in the shape of micro-/nano-structures [30,39,40].

A piezoresistive pressure sensor using a typical micro-/nano-structure was fabricated [39]. The developed device has a structure comprising a silicon cavity, a thin PLGA membrane, and a strain gauge. When pressure is applied, the PLGA membrane is bent into the Si substrate cavity. At this time, strain is applied to the Si nanomembrane strain gauge in the PLGA membrane, and the resistance changes. In order to maximize the sensitivity, the Si nanomembrane (NM) was placed on the clamp of the PLGA. To implement the proposed idea, a device was fabricated using microfabrication and transfer methods [39]. To characterize the piezoresistive performance of the device, the device was placed in a syringe filled with artificial cerebrospinal fluid (CSF) at physiological temperature (37 °C). The pressure was manually controlled by moving the plunger part of the syringe with the hands. The sensitivity of the device was 83 Ω/mmHg (622.55 Ω/kPa^−1^) [39]. The biodegradability of the device was also tested by placing the device in a poly(dimethylsiloxane) (PDMS) chamber filled with an aqueous buffer solution of pH 12 at room temperature. In the test, the Si NM and SiO_2_ components dissolved within 15 h, and the nanoporous silicon disappeared within 30 h [39].

Although the research on the piezoresistive pressure sensor using micro-/nano-structures greatly improved the performance, it is also necessary to improve the lifetime of the device in order to be able to use these devices in bioimplantable applications. When actual biofluids come into contact with active materials, water permeates through the material of the device or the interface between materials which leads to the deterioration in the performance of the device. Therefore, these functional lifetime issues must be resolved to meet clinical needs. Initially, passive polymer layers were added to delay the time during which the biofluids and the active area are in contact. However, biodegradable polymers such as PLGA have the limitations of poor stability because the hydrophilic nature of the material causes swelling and water permeation, which causes premature fracture, buckling and/or dissolution in materials in the active area. In a more recent study, a method that uses a thermally grown silicon dioxide (t-SiO_2_) layer was reported. A t-SiO_2_ layer has a low decomposition rate (rates of several hundredths of a nanometre per day) and can serve as a defect-free biofluid barrier in a wide area, resulting in a longer functional lifetime. A piezoresistive silicon pressure sensor with t-SiO_2_ as encapsulation was facbricated using a silicon on insulator (SOI) wafer (Figure 6d) [30]. Since the developed device also consisted of an Si NM resistor and a Si cavity similar to the previous device, the pressure could be measured through the change in the resistance of the Si NM. In vitro evaluations that mimic the thermodynamic conditions inside (artificial CSF; pH 7.4 at 37 °C (physiological temperature)) the device show that comparison of the voltage responses of the sensor with the measured pressures of a commercial sensor reveal a linear correspondence (Figure 6e) [30]. In the reliability test, the functional lifetime of the device was maintained up to 22 days, and the sensitivity showed a reliability of ±1.5% (Figure 6f) [30].

Although the t-SiO_2_ can increase the functional lifetime by encapsulating the device, the rate of degradation of the device is very slow (10^−3^–10^−1^ nm/day) [40]. It has a timespan in years which is much longer than clinically relevant operational requirements required to achieve complete dissolution of the t-SiO_2_ layer. Therefore, the researchers conducted a study to reduce the physical lifetime of the device, while continuing to have the required functional lifetime, and an encapsulation method combining the thin, lightly doped micromembrane of monocrystalline Si (Si MM) and natural wax was developed. The lightly doped Si MM is impermeable to water and is very thin; thus, the functional lifetime is improved, and the physical lifetime is also relatively shortened. Therefore, a piezoresistive pressure sensor, taking the different lifetimes into consideration, was fabricated. This device is mechanically stable and its lifetime is appropriately controlled using an Si MM encapsulation on the pressure-sensing part and covering the other part with natural wax [40]. The fabrication method for the device begins with fabricating the lightly doped Si MM that acts as a biofluid barrier. Then, using the conventional semiconductor processes, such as photolithography and etching, the Si MM, Si NM, and Mg substrate are patterened, and the components are assembled with the PLGA. Finally, the surface of device excepted the sensing area is covered with wax for the device-encapsulation. For the encapsulation, a natural wax material, with excellent lifetime, using Beeswax (CB01) and Candelilla wax (CB10) was also developed. Various ratios of the wax mixtures were tested to identify the optimal natural wax for encapsulation, and CB32 (ratio of Candelilla wax to Beeswax: 3:2) was identified as the best as it prevented water from penetrating into the device for more than 22 days [40]. Another advantage of the developed sensor is that there is little change in characteristics due to device biodegradation. Earlier pressure sensors based on a similar principle showed a change in the membrane thickness, due to degradation of the encapsulation layer with time, which results in a performance change such as a change in sensitivity. However, the fabricated device was designed considering a mechanical neutral line, minimizing the change in performance resulting from degradation of the encapsulation layer [40].

#### 3.2.2. Biodegradable Piezo-Capacitive Pressure Sensor

The Bao group focused on a capacitive biodegradable pressure sensor that exploits the microstructured dielectric material. The material used as the dielectric was PGS. In general, elastomers having a smaller mechanical modulus exhibit greater viscoelasticity; thus, their mechanical response is relatively slow. However, PGS has a relatively high mechanical modulus of ≈1 MPa, and the viscoelasticity can also be controlled during the manufacturing processes, such as through reaction, curing temperature and time. The microstructure which is a square pyramidal structure of the PGS helps to achieve a high performance as a pressure sensor. This square pyramidal structure of the PGS has a large mechanical deformation and the fast response time because of the structure having an air layer inside [47,118,119,128,129,130]. In practice, the fabricated device shows a high sensitivity and fast response time. In the pressure ranges of p < 2 kPa and 2 kPa < p < 10 kPa, the sensitivity was 0.76 ± 0.14 kPa^−1^ and 0.11 ± 0.07 kPa^−1^, respectively. Moreover, the response time was approximately 100–200 ms [47]. In vitro, the biodegradation studies of the device were also performed for 7 weeks in an incubation filled with PBS solution (pH 7.4) at 37 °C. The device consisted of PGS (dielectric layer), polyhydroxybutyrate/polyhydroxyvalerate (PHB/PHV, substrate layer), PVA (Adhesive layer), and iron-magnesium (Fe-Mg, electrode). After 7 weeks, the weight of the device remains at about 85% of its initial value. Fe-Mg of the device was dissolved for the first time, but both PGS films and PHB/PHV was not fully dissolved. These materials require a duration of at least a few months to fully dissolve [47].

A two-axis pressure sensor integrating pressure and strain and a simple pressure sensor was developed using biodegradable material. The sensor was designed to measure pressure in the vertical direction and the strain in the horizontal direction to distinguish between the two physical stimuli, independently. The developed two-axis pressure sensor also employs a capacitive mechanism. The pressure sensing and strain sensing was performed by measuring the capacitance between two electrodes while varying the distance between two different electrodes and by varying the overlapped area, respectively (Figure 7a) [131]. To fabricate the proposed two-axis sensor, a pressure sensor and a strain sensor are stacked together and then combined using a UV-curable, biodegradable polymer (poly (octamethylene maleate (anhydride) citrate); POMaC) [132]. The pressure sensor consists of a square pyramidal microstructure of PGS and a pair of Mg electrodes to detect pressure with high sensitivity. The varying overlapped electrode areas for the strain sensor were fabricated by designing separate thin-film comb electrodes with soft polymer. Figure 7b shows the strain response of the fabricated device for five consecutive cycles. The applied strain is 0–15%, and the capacitance of the device changes between 3.5 and 7 pF [131]. The device shows negligible hysteresis. Figure 7c shows the pressure response characteristics for six consecutive linear loading–unloading cycles. In the pressure range from 0 to 100 kPa, the capacitance of the device changes between 3.5 and 11.5 pF, and the device shows the negligible hysteresis [131].

In addition to the microstructured dielectric approach, there is also a method to construct the nanostructured dielectric layer using composite nanofiber membranes (CNMs). A typical method for fabricating the CNMs is the electrospinning process. The electrospinning process is to continuously draw out fibers from a viscous polymeric solution through rapid solvent evaporation using an electric field. The advantage of this approach is that it can produce fibers with very thin diameters of a few micrometers or nanometers. In addition, it has the advantages of excellent flexibility, porosity, lightweight and compatibility with the printing process. This method has been actively used to fabricate the active layer of the strain sensor and the dielectric layer of the pressure sensor. CNMs using biodegradable composite polymer (poly(lactic-co-glycolic acid)-poly(caprolactone); PLGA-PCL) as a dielectric material for the biodegradable pressure sensor was reported (Figure 7d) [133]. CNMs can have very low mechanical modulus because they have a considerable number of air pores owing to the fiber network inside (Figure 7e) [133]. The fabricated sensor has a sensitivity of 0.863 ± 0.025 kPa^−1^ in a specific pressure range (0 < *p* ≤ 1.86 kPa) and high detectivity (1.24 Pa at 10 mgf) (Figure 7f) [133]. In addition, this sensor has a low pressure-detection value of 1.24 Pa, and good average response and recovery times of 251 ms and 170 ms, respectively. In vitro the biodegradation studies of the device were also performed for 18 days in an incubation filled with PBS solution at 37 °C. After 14 days, the weight of the device remains about 40% of its initial weight. The device continued to degrade until 18 days [133].

Thanks to the emergence of micro-/nanofabrication, various biodegradable materials have come to be fabricated with high designability and scalability, and it enables the development of wireless biodegradable pressure sensors exploiting various passive biodegradable components. The biodegradable wireless pressure sensor can be fabricated by combining a variable capacitive sensor (C) and an inductor (L). This LC combination will have a resonance frequency, and the change in resonance frequency caused by an external stimulus can be measured by an external secondary coil connected to the impedance analyzer wirelessly. A wireless pressure sensor was fabricated by building a simple LC circuit using microstructured biodegradable materials, metal alloys (iron-zinc alloy; Fe-Zn) and polymers (polylactic acid; PLA, polycaprolactone; PCL) [64]. The Fe-Zn alloy was designed to promote the corrosion of Zn. This is because pure Zn decomposes slowly in a biological environment. The microfabricated coil acts as an inductor, and the two microplates and air cavity act as a capacitor. The principle of wireless measurement of the sensor is that when pressure is applied to the sensor, the distance between the plates decreases, so the capacitance value changes. The phase and magnitude of the RF signal with respect to the resonant frequency can be measured [64]. The performance of the fabricated sensor was evaluated in a sealed chamber. The resonant frequency signal of the sensor, which depends on the applied pressure, was measured in wireless manner in air and in a saline (0.9% NaCl in deionized water) environment. When the applied pressure was increased, the resonant frequency significantly decreased. In the pressure range of 0–20 kPa, the resonant frequency shift of the device is −39 MHz/kPa and −35 kHz/kPa in air and saline, respectively [64]. In vitro the biodegradation studies of the Fe-Zn were also performed for 300 h in 0.9% saline (NaCl, pH 6.8) at 37 ± 0.5 °C. The degradation rate was divided 3 stages. The degradation rate of stage 1 (between 0 and 72 h), 2 (between 84 and 180 h), and 3 (after 200 h) was 0.15 mg/h, 0.05–0.1 mg/h, and under 0.04 mg/h, respectively [64].

Recently, the functional lifetime of a biodegradable wireless pressure sensor has been extensively studied. Lu et al. attempted to increase the functional lifetime by encapsulating the device with silicon nitride (Si_3_N_4_) micromembranes and natural wax in the wireless sensor [135]. The thin micromembranes (2 μm), Si_3_N_4_, help to use the mechanical properties of the flexible top electrode. In practical, the bioresorbable water barrier structures, Si_3_N_4_ micromembranes and natural wax, prevent water permeation and ensure stable operation electrode [135]. The rate of degradation of the Si_3_N_4_ and wax are 4.5–30 nm per month and 10 μm per month, respectively. In addition, there were attempts to reduce the sensor’s parasitic capacitance (C_p_) and increase the sensitivity as well as the functional lifetime of the sensor. The sensor was designed by removing a part of the bottom electrode from the existing parallel plate capacitor structure, creating an additional trench around the cavity, and adding wax to the edge between the two electrodes. This design reduced the C_p_ by relaxing the overlap between the top and bottom electrodes, which is one of the main sources of C_p_, and increased the sensitivity by increasing the deformation. A sensor designed in this manner can reduce noise compared to the capacitance sensor built with an intuitive approach. As a result of measuring the applied pressure and the resonant frequency over time, using the external coil in vitro environment, the performance of the sensor was found to be similar to the performance of a clinical standard device used for measuring intracranial pressure (ICP) monitoring electrode [135]. In vitro the biodegradation studies of the device were also performed for 44 days in PBS solution at 37 °C. The device consisted of Mg, PLGA, Si_3_N_4_, Zn, and wax. Over a period of 44 days, the device was gradually dissolved. After 44 days, leaving only the S_i3_N_4_ and the wax. The degradation rate of Si_3_N_4_ and wax are 4.5–30 nm/month and 10 μm/month, respectively. The full degradation of the device materials will require a duration of at least a few months [135].

Recently, a cuff-type biodegradable wireless piezocapacitive pressure sensor having high flexibility has been studied (Figure 7g) [134]. The sensor consists of a capacitive pressure sensor, which detects the applied pressure by observing changes in the electrical fringe field. Owing to the microstructure of pyramidal dielectric, the device deformation causing the fringe field, and the change in capacitance are easily generated, when an object comes into contact and a pressure is applied. The sensor was also made with soft encapsulation material, a biodegradable polymer (POMaC) and comb-designed metal electrode, so that the device could have a flexible configuration (Figure 7h). In particular, the developed sensor was designed to have a micro-bilayer structure as the coil (Figure 7i). The micro-bilayer coil structure is favorable to obtain a lower resonant frequency owing to the high mutual inductance. This causes the signal to undergo a lower attenuation within the body, resulting in a longer operating distance of the wireless sensor. Moreover, this design can provide a larger frequency shift for a given applied pressure, which means higher sensitivity. The device was made entirely of biodegradable materials, including PGS, POMaC, PHB/PHV, PLLA, and Mg [134].

#### 3.2.3. Biodegradable Piezoelectric Pressure Sensor

With the recent development of piezoelectric biodegradable materials, it has become possible to realize biodegradable piezoelectric pressure sensors. A piezoelectric pressure sensor based on β-glycine-chitosan (β-Gly/CS) microfilm, a biodegradale organic material, with a noncentrosymmetric polar structure was fabricated (Figure 8a) [136]. This β-Gly/CS microfilm was fabricated by embedding grown bio-organic glycine crystals (β-glycine) inside a chitosan polymer used as a matrix material (i.e., self-assembly). The fabrication method using the self-assembly makes the unstable β-glycine with a piezoelectric coefficient (d_16_ = 174 pmV^−1^) uniform and thermodynamically stable, making it possible to act as a piezoelectric film (Figure 8b) [136]. To fabricate the piezoelectric pressure sensor, a simple solvent-casting method is employed. The sensor was fabricated by depositing Au electrode using a hard mask on both sides of the β-Gly/CS microfilm fabricated earlier. The fabricated piezoelectric pressure sensor was characterized using a vibration system (S50018) to measure the performance of the output voltage of the device. Under dynamic vibration, the device stably generated the open circuit output voltage with a fast response time (<100 ms). In particular, the measured sensitivity of the sensor was 2.82 ± 0.2 mV/kPa in the pressure range of 5–60 kPa (Figure 8c).

To increase the sensitivity in a specific pressure range of the biodegradable piezoelectric pressure sensors, Curry et al. fabricated a biodegradable piezoelectric pressure sensor based on stretched poly(L-lactic acid) (PLLA) microfilm (Figure 8d) [122]. Stretched PLLA microfilm was fabricated using PLLA, a biodegradable polymer approved by the Food and Drug Administration (FDA) [137,138]. In particular, they developed the high performance PLLA piezoelectric film through thermal annealing, mechanical stretching, and cutting processes. Thermal annealing and mechanical stretching of the PLLA microfilm helped form the piezoelectric property in the PLLA film because these methods could create a 45° alignment of the carbon-oxygen double bonds (C=O) that intrinsically have the piezoelectric property [139,140]. Then, by cutting the film with a 45° angle along the stretching direction, the optimized PLLA piezoelectric film is fabricated. To complete the fabrication of the piezoelectric pressure sensor, Fe and Mg alloy electrode is deposited on both sides of the PLLA microfilm using an electron beam evaporator. Then, Mo wires are placed on each surface of the electrode and it is encapsulated within the PLA layers. Finally, it was enclosed using biodegradable PLLA glue and a thermal bag sealer. Figure 8e shows the optical image of the fabricated device. Figure 8f shows that the output voltage increased as the applied pressure was increased from 0 kPa to 18 kPa. The evaluated sensitivity was about 0.12 V/kPa in the range of 0 kPa < *p* < 2 kPa, and a sensitivity is about 0.013 V/kPa in the range of 2 kPa < *p* < 18 kPa [122]. In vitro the biodegradation studies of the device were also performed in the buffered solution at an accelerated-degradation temperature of 74 °C. The device consisted of biodegradable materials (PLA [141], Mo [142], and PLLA), and completely degraded after 56 days [122].

To improve the piezoelectric response of PLLA, the same research group attempted to fabricate a PLLA nanofiber film (Figure 8g) [123]. The PLLA nanofiber film was made of nanofiber PLLA material using a rotating drum. The rotating drum aligned the crystal domains of the PLLA nanofibers and improved the piezoelectric properties because of the uniform and scalable unidirectional polarized structure that was created. Figure 8h shows the orientation of the crystal domains inside 4000 rpm electrospun PLLA nanofibers, and the inset shows the flexibility of the sensor. Moreover, Figure 8i shows a 1.8 times higher piezoelectric response of the PLLA nanofibers film compared to the stretched PLLA microfilm.

#### 3.2.4. Biodegradable Optical Pressure Sensor

A measurement method using resonant peak positions in the reflection spectra is one of the representative methods of biodegradable optical pressure sensors. Shin et al. fabricated a biodegradable optical micro-pressure sensor using the principle of Fabry–Perot interferometers (FBI) (Figure 9a) [143,144,145,146]. To measure the pressure using the sensor, a tunable laser source illuminates the sensor through a PLGA optical fiber. A photodetector measures the resonator signals corresponding to the thickness of Si slab microcavity, the volume of which changes depending on the applied pressure (Figure 9b) [143]. The amorphous silica layer (thickness: ~200 nm), the Si slab microcavity (thickness: 10 μm), and t-SiO_2_ layer (thickness: ~250 nm) serve as glass, which can reflect the light, a resonator, and a passivation film, respectively. For in vitro evaluations, the FPI based biodegradable pressure sensor is immersed in PBS (pH 7.4). In the PBS solution, the device shows a sensitivity of −3.8 nm/mmHg with an accuracy of ±0.40 mmHg in the pressure range of 0–15 mmHg (Figure 9c) [143]. The lifetime of the device is also evaluated. For the evaluation of the functional lifetime, the sensor was immersed in PBS at 37 °C for 8 days. The device did not show any significant degradation in performance.

The same group further fabricated a sensor that measures pressure using microcavities through photonic crystal structures (PCs) [125,147,148,149] that is a periodic structure in the nanometer range (Figure 9d) [143]. The sensor with nanostructures of PCs on a flexible diaphragm yields sharp resonance peaks with a high Q factor. When pressure is applied to the sensor, the size of the nanostructure changes, and then the resonance peak shifts. The nanostructures of PCs also make wireless measurements possible, it can reduce the possibility of problems such as infection [143]. The principle of this pressure sensor is that when the pressure increases, the grating size of the PC microcavities in SI NM increases; the grating size of PC microcavities in SI NM increases with the increased pressure, resulting in the increase in the resonant wavelength. The fabrication process of the sensor consists of fabricating SOI-A with PC microcavities and SOI-B with silicon trench via e-beam lithography and deep reactive-ion etching (DRIE), respectively, and combining with PDMS between SOI-A and SOI-B (Figure 9e). In particular, PDMS is transformed into silica by calcination [143]. In in vitro evaluation, the environment has a tunable laser source, tilt stage (chamber), circulator, and photodetector. The sensor has a sensitivity of 1.9 pm/mmHg and an accuracy of ±0.64 mmHg in the wide range of pressure (0 to 100 mmHg) (Figure 9f) [143].

### 3.3. Biomedical Applications

Attempts have been made to apply the developed pressure sensors to practical biomedical applications. This section introduces the biomedical applications using biodegradable pressure sensor such as ICP, blood pressure, tendon, and intra-abdominal pressure.

#### 3.3.1. Application for Intracranial Pressure (ICP) Monitoring

In the body, ICP monitoring is necessary to diagnose diseases such as traumatic brain injury, aneurysms, brain tumors, hydrocephalus, stroke, and meningitis. Tests on animals (rat model), using developed microstructured Si piezoresistive biodegradable pressure sensor, were conducted in Section 3.2.1 [39]. In the rat model, the initial piezoresistive-type biodegradable pressure sensor was connected to a wireless transmitter with a potentiostat through a degradable wire [39]. The wireless transmitter was used to measure pressure after surgery; it was protected using a head protector, allowing the rat to freely move around [39]. In in vivo evaluation, to measure ICP, the sensor was mounted on the top of the skull. A comparative test of the clinical intracranial pressure sensor (Integra Life Sciences, Princeton, HJ, USA) transplanted in parallel with a biodegradable pressure sensor encapsulated with polyanhydride showed stable results for three days [39]. Further, in vivo observation of the changes in ICP as a function of time in the Trendelenburg and reverse Trendelenburg positions were made. Trendelenburg position and reverse Trendelenburg position are associated with accumulation and depletion of blood in the brain, respectively. ICP increases in Trendelenburg position (30° head-down) as compared with the supine position, and decreases in reverse Trendelenburg (30° head-up) [39].

Biodegradable resistive pressure sensors focused on the encapsulation (t-SiO_2_ nanolayer for device-protection) in Section 3.2.1 also showed highly stable results in the in vivo test, using a male Lewis rat model [30]. For the ICP monitoring, a flank squeeze (contract, release), Trendelenburg position (flat, 30° head-up, down), and a mannitol infusion (dose: 2 g per kg weight) were performed. In the flank squeeze test, the ICP increased up to 20 mmHg in the contract, and decreases below 10 mmHg in the release. In the Trendelenburg position test, the ICP decreased below 7 mmHg in the head-up, and increased above 9 mmHg in the head-down. For mannitol infusion in the saphenous vein, the ICP decreased from 8 to 1 mmHg after intravenous drug infusion. In addition, the fabricated sensor device also collected the varying ICP continuously for 18 days. The device shows the stable functional lifetime as a signal at 18 days after surgery (red: bioresorbable sensor; blue: commercial sensor). Over a period of 25 days, the response of the sensor showed an absolute accuracy within ±2 mmHg, baseline drift within ±1 mmHg, and negative drift of ~3 mmHg [30]. 

The biodegradable wax-encapsulated piezoresistive pressure sensor was also tested for ICP measurement [40]. The procedure for implantation consisted of exposing the skull area, opening a craniotomy defect, attaching a sensor using dental cement over the defect and commercial bioresorbable glue (COSEAL surgical sealant) on the edge, and sealing the cavity. At this time, the opening of the skull ensures contact with the cerebrospinal fluid (CSF) to enable ICP monitoring. The in vivo sensor was tested for Trendelenburg position (30° head-down) and reverse Trendelenburg position (30° head-up), compressing and releasing the rat’s flank for a short period (<3 min). Baseline and sensitivity drift of the sensor through the flank contracting and releasing was measured on 0, 7, 14, and 21 days postsurgery. Baseline and sensitivity showed a variation of ±1.0 mmHg and 2.1% of 15.0 Ω mmHg^−1^ over 3 weeks, respectively [40].

In addition to the resistive type, there was also an in vivo evaluation of the rat model for the optical FPI type biodegradable pressure sensor (Figure 9a and Figure 10a) [143]. The implantation procedure, which was approved by the Institutional Animal Care and Use Committee (IACUC) of Northwestern University, consisted of drilling a small defect hole inside the skull, and implanting the sensor inside, putting a film of PLGA with a hole in the center (dimensions, 5 mm × 5 mm × 10 μm; hole diameter, ~400 μm) on the top, and bonding the PLGA film and the skull by applying a layer of bioresorbable glue to form an airtight seal on the intracranial cavity. The hardened glue helps to prevent changes in the sensor–fiber alignment due to shear or slanting effects during and after implantation (Figure 10b) [143]. In in vivo testing, the sensor measured the ICP by squeezing and holding the rat’s flank. Figure 10c shows the optical spectra and pressure calibration curves obtained from the FPI pressure sensor. For contracting the flank, the sensor’s ICP was measured in the range 3–13 mm Hg, and the sensitivity was 3.1 nm/mmHg [143].

A study on the biodegradable wireless sensor for ICP monitoring was also conducted. The biodegradable wireless piezocapacitive pressure sensor with wax encapsulation, including an LC circuit (primary coil), was tested in vivo [135]. The implantation procedure includes sterilization of the head, creation of craniectomy over the head, and implantation of a bioresorbable sensor over the skull. Applying dental cement (Fusio Liquid Dentin) and curing under ultraviolet light fixed the implant on the skull and made the sensor airtight [135]. In the animal test, the ICP was increased by 5–10 mmHg for 15 s by gripping the rat’s flank by hand, under anesthesia, and then released. When measuring the ICP, the body temperature of the rat measured using a rectal probe was 35.5 °C. A result of the ICP monitoring in rats for 4 days after implantation shows a stable functional lifetime. The readout system consists of a single turn coil (external secondary coil) connected to an Agilent E5062A or Agilent portable N9923A vector network analyzer (VNA), and the real and imaginary parts of the S-matrix element S_11_ were measured by setting the VNA to reflective mode. At this time, the variable resonance frequency of the LC circuit was measured between 309 and 312 MHz [135]. 

#### 3.3.2. Application for Pressure of Blood Vessels, Tendon, and Intra-Abdominal

In addition to the ICP, the biodegradable pressure sensor can be utilized to measure the pressure of other tissues, for various in vivo applications, such as the pressure of blood vessels, tendon, and intra-abdomina. Cardiovascular diseases are the leading cause of premature death worldwide today, and hypertension has been defined as the main risk factor and leading cause of cardiovascular diseases. It is important to detect hypertension and continue monitoring before too much damage occurs. Measurement of the carotid-femoral pulse wave velocity (cf-PWV) is currently the “gold standard” for measuring aortic arterial stiffness which is directly related to hypertension, and is a reference in the international cardiology/hypertension treatment guidelines [27]. A biodegradable piezocapacitive pressure sensor with a pyramid microstructured dielectric is suitable for cardiovascular monitoring such as arterial tonometry and cf-PWV measurement due to its high sensitivity and fast response time [47]. To measure the cf-PWV, the arterial pulse wave and an electrocardiogram (ECG) were recorded simultaneously for time reference. At this time, the sensor was first attached to the adult’s neck (carotid artery) and then to the groin (femoral artery). The cf-PWV (v) was measured by dividing the time delay (t) between the feet of the carotid and the femoral pulse signals by the distance (d) between the neck and the groin. The measured cf-PWV is 7.5, a result typically obtained in healthy subjects [47].

After injury to the tendons, ligaments, and joints, body tissues such as the hard tissues (bones) and soft tissues (tendon, skin, muscle) undergo changes in their native biomechanical properties to repair themselves. The purpose of surgery and rehabilitation is to restore tissues to their pre-injury function. Monitoring the biomechanical characteristics of the repair site in real time can be a diagnostic tool that can predict the healing process for personalized rehabilitation. The level of strain and strain rate of tissues during the rehabilitation are the most important parameters that can characterize the biomechanical properties and healing stage of soft tissues. An implantable sensor should be able to measure the typical tendon strains (<10%) without interfering with the natural movement of the tendon. In addition, it should be possible to measure the pressure applied to the repaired area that directly affects the healing profile [131]. The previously developed sensor (Figure 7a) that can measure strain and pressure independently without interfering with each other is suitable for this biomedical engineering application [131]. For in vivo testing, three Sprague Dawley rats were treated in accordance with the regulation of the animal care and use committee of Veteran Affair palo Alto Health Care System Research Administration. The sensor was implanted into the subcutaneous paravertebral pocket under isoflurane inhalation anaesthesia (Figure 11a). Figure 11b shows the signals of pressure (top) and strain (bottom) successfully recorded 3.5 weeks after sensor insertion for pressure and strain detection. The strain and pressure were measured using a E4980A Agilent Precision LCR meter to measure the sensor capacitance in the vertical and horizontal directions, respectively. The sensor could distinguish between strains as small as 0.4% and the pressure exerted by the salt particles (12 Pa) without interfering with each other [131].

Failure after reconstructive surgery, including microsurgical anastomosis, may occur due to the formation of haematoma or thrombosis in the artery or vein. To ensure successful recovery after the surgery, it is important to measure blood vessel movement. Therefore, it is important to detect failure such as anastomosis early by monitoring the surgical site. Using the developed flexible cuff-type biodegradable piezocapacitive pressure sensor (Figure 7g), an in vivo testing was performed to wirelessly observe the movement of blood vessels [134]. This sensor measured the pulse rate of the femoral artery in Sprague Dawley rats (300–350 g, male; ENVIGO) in compliance with the regulation of the animal care and use committee of Veteran Affair Palo Alto Health Care System Research Administration.

The implantation procedure was performed under isoflurane inhalation anaesthesia. The sensor was wrapped around the femoral vessel and mounted on the abductor muscles with sutures. As we mentioned in the above section, this biodegradable sensor device can detect the vessel movement wirelessly. For the wireless sensor testing, the coil structure of the device was placed on a groin fat pad. Wireless measurement is performed by inductively coupling the implanted sensor coil structure with an external reader coil connected to a vector network analyzer. The reader coil measures the $\Delta f_{0}$ value of the LCR resonator through the scattering parameter S11. A pulse rate of 3.47 beat per second (b.p.s) was recorded (Figure 11c). Using this sensor, an occlusion test (femoral artery was blocked for 1 min and then released) was further performed; this test mimicked early clot formation in the vessel by placing two nylon sutures in the femoral artery on either side of the sensor and applying tension (which slows the blood flow through the artery and reduces the extension of the artery diameter). From the experimental results, it was clearly confirmed that the blood vessels have different movements during tension and release, and a pulse rate of 4.29 b.p.s was recorded (Figure 11d) [134].

It was confirmed that the piezoelectric pressure sensor with the stretched PLLA could detect the breathing pattern by measuring intra-abdominal pressure [122]. The in vivo test was performed under the approval of the Connecticut Health Center’s IACUC. A sensor coated with a very thin medical glue is inserted into a small incision (8 mm) made under the diaphragm of the rat in the abdomen. Small Mo/PLA wires from the sensing patch were connected to an external charge amplifier circuit connected to an oscilloscope, through a sutured wound, to measure the voltage. The respiration of rats under normal anesthesia after resting them for 15 min after surgery was monitored. Signals generated in the mice were completely suppressed after euthanasia due to an overdose of anesthetics. When the signal was alive, a frequency of ~2 Hz and ~0.1 N/cm^2^ (~1 kPa) were measured. The sensor also detected abnormal respiration with a lower frequency and greater pressure until the animal died after an anesthetic overdose [122].

#### 3.3.3. Biocompatibility

Biocompatibility can be defined as the compatibility between a material and biological system [150]. Therefore, biocompatibility of the devices through all stages of the life cycle is essential. In previous studies of biodegradable pressure sensors, immunohistochemistry (IHC), complete blood count (CBC) and blood chemistry, and histopathology studies were conducted. Immunohistochemistry is an immunostaining method that uses the reaction between antigen and antibody to identify substances present in cells or tissues. After implantation of the microstructured Si piezoresistive type sensor, comprehensive studies on the IHC of brain tissues several times (2, 4 and 8 weeks) prove that the by-products of the sensor and dissolved device in the space within the skull are biocompatible through the absence of an inflammatory response in confocal fluorescence microscopy (CFM) images [39]. CFM images were double-immunostained for glial fibrillary acidic protein (GFAP) (red) to detect astrocytes and ionized calcium-binding adapter molecule 1 (Iba1) (green) to identify microglia/macrophages (dashed line is the implant site). This indicates that there is no concentrated aggregation of glial cells at the implantation site for all time ranges and no apparent response of brain glial cells. The astrocytosis and microglial activity on the cortical surface are within the normal range [39].

The CBC is a blood test used to evaluate overall health and detect a wide range of disorders. A blood chemistry test is a blood test that measures specific chemical content in a sample of blood. This can help make sure that certain organs are working well and locate disorders if any. Figure 12a shows the results of CBC and blood chemistry for the biodegradable resistive pressure sensor protected with t-SiO_2_ nanolayer (Figure 6d) [30]. The CBC (left in Figure 12a) shows no significant difference between implanted and control mice over five weeks in the average counts of white blood cells (WBC), red blood cells (RBC), platelets (PLT), and in the levels of haemoglobin (HGB) and hematocrit (HCT). Blood chemistry (right in Figure 12a) shows that the blood levels of enzymes and electrolytes fall within the confidence intervals of control values. Normal levels of alanine aminotransferase (ALT), cholesterol (CHOL), triglyceride (TRIG), phosphorus (PHOS), blood urea nitrogen (BUN), glucose (GLU), calcium (CAL), albumin (ALB), and total proteins (TP) indicates the absence of disorders in the liver, heart, kidney, bone and nerve and good overall health. Control data were provided from a mouse supplier (grey) or collected from 22–24 mice (cyan) [30].

Histopathology is the diagnosis and study of diseases of the tissues, and involves examining tissues and/or cells under a microscope. Histopathologic evaluation of tissues obtained from a control mouse and a mouse implanted with the device protected with the t-SiO_2_ nanolayer (Figure 6d) for five weeks showed absence of an inflammatory response, ischaemia/tissue necrosis, and architectural/histologic abnormalities indicating no significant changes within major organs such as the brain, spleen, heart and kidney (Figure 12b) [30,151].

## 4. Summary and Outlook

This paper highlights the recent advances in micro-/nano-structured biodegradable pressure sensor devices through discussions on material options, sensing mechanisms, device design, biocompatibility aspects, and biomedical applications. Various classes of materials including conductors, semiconductors, and insulators are summarized. The materials are categorized into wet and dry transient materials based on certain aspects in the biodegradation mechanisms; the materials are, generally, degraded based on hydrolysis in wet conditions, but a few materials can be spontaneously degraded in dry conditions or when triggered by a certain stimulus, resulting in wider material selection and utilization at the device level. Pressure sensing mechanisms, including piezoresistive, piezocapacitive, piezoelectric, and optical, were also introduced. The general theory of each mechanism is not only reviewed, but the structural strategy to improve device performances, such as sensitivity, response time, stability, and reliability, is also discussed in detail. An in depth review of micro-/nano-structured biodegradable pressure sensors that demonstrates their excellent pressure sensing performances, based on physical phenomena induced by the associated miniaturized structure is presented (Table 1).

We discussed the recently developed micro-/nano-structured biodegradable pressure sensors including their concept, fabrication, and performances. Finally, we also introduced real cases showing the feasibility of the developed micro-/nano-structured biodegradable pressure sensors for diagnosis of disease and healthcare in wearable and implantable biomedical applications without any biocompatibility issues. Despite of the recent development in biodegradable pressure sensors, there are still some issues, such as sensitivity, operating range, and controllability of biodegradability, essentially to be improved for accurate pressure monitoring of various tissues and organs. In other words, regarding there are huge recent demands for healthcare and diagnosis of diseases in early stage through the pressure sensing of diverse tissues and organs, such as intracranial pressure [152,153,154,155,156,157], intraocular pressure [158,159,160,161,162,163], heartbeat [164,165,166], and bladder pressure [167,168,169,170], we expect that the high-performance biodegradable pressure sensors will be much more important for diverse biomedical applications in the future.

## Figures and Tables

**Figure 1 biosensors-12-00952-f001:**
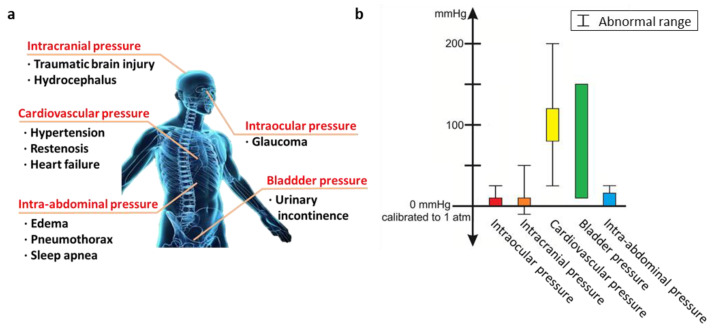
Pressure monitoring. (**a**) Pressure is linked to several diseases in various organs of the body [24]. (**b**) Relevant pressure ranges for in vivo pressure monitoring for diagnostic applications [29].

**Figure 2 biosensors-12-00952-f002:**
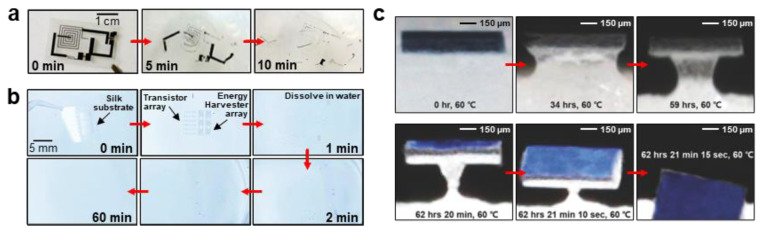
Examples of transient electronics. Red arrows indicate the elapse of time. (**a**) Optical images of Si diodes, Si/MgO/Mg transistors, Mg/MgO inductors and capacitors, and Mg resistors and interconnects on a silk substrate [32]. (**b**) Photographs of ZnO thin film transistor arrays and mechanical energy harvester arrays on silk substrate [41]. (**c**) Optical images of an array of solar cells on a dry transient CDD substrate [42].

**Figure 3 biosensors-12-00952-f003:**
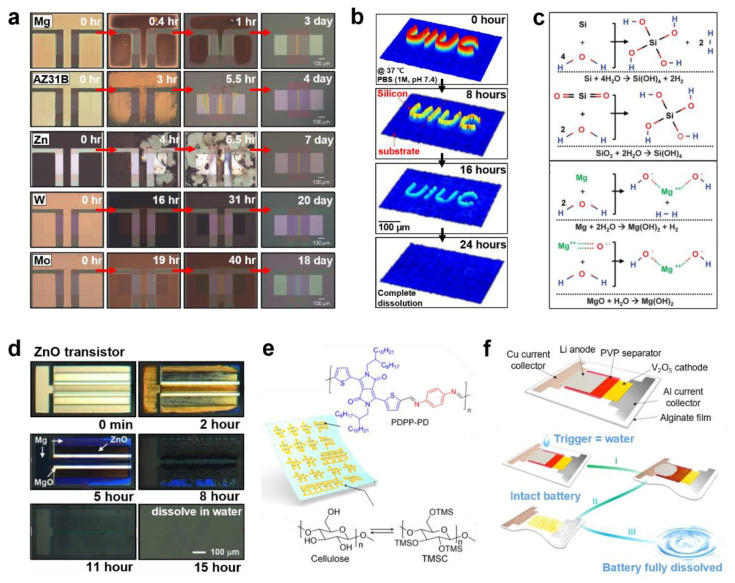
Examples of wet transient materials. (**a**) Optical images of metal dissolution behavior [43] (Mg, AZ31B Mg alloy, Zn, W, and Mo). Red arrows indicate the elapse of time. (**b**) Transmission-mode laser diffraction phase microscopic (DPM) image of Si NMs (∼100 nm-thick) at various times of immersion in phosphate-buffered solution (1 M, pH 7.4) at physiological temperature (37 °C) [61]. Black arrows indicate the elapse of time. (**c**) The chemical reaction for degrading mechanisms of Mg and Si in water [32]. (**d**) Optical images of a ZnO thin film transistor at various instants during the dissolution consisting of ZnO (active materials), Mg (electrodes, contacts, and interconnects), and MgO (gate and interlayer dielectrics) [41]. (**e**) Illustration and structures of biodegradable polymer (PDPP-PD) semiconductors on biodegradable cellulose substrate [49]. (**f**) Schematic process of water-triggered transient battery dissolution [62].

**Figure 4 biosensors-12-00952-f004:**
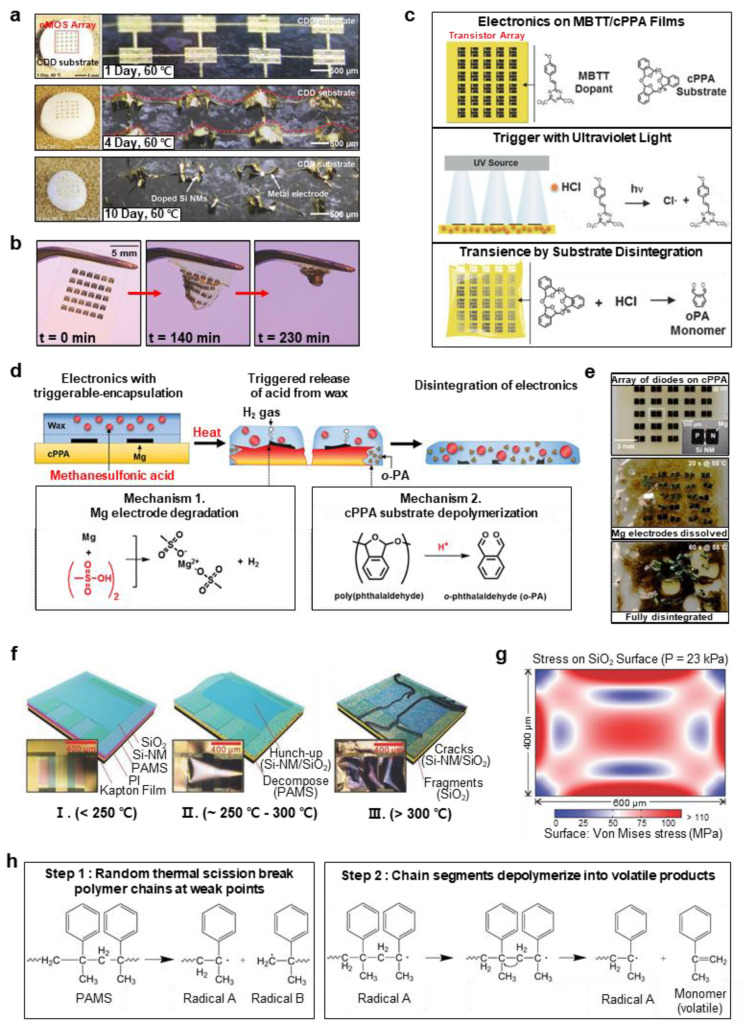
Examples of dry transient materials. (**a**) Microscopic images showing the time sequence of disintegration behavior of an array of transistors due to a sublimation of the supporting substrate of CDD [42]. Optical images (**b**) and mechanism with chemical structure (**c**) of UV light-triggered transient material [93]. (**d**) Illustrations and chemical structures of the disintegration process, and (**e**) optical images of the transient behavior of 55 °C heat-triggered transient device [94]. (**f**) Schematics of device structure failure and (**g**) FEA analysis image of stress distribution on SiO_2_ layer surface of 330 °C heat-triggered transient device. (**h**) Degradation mechanism of PAMS [95].

**Figure 5 biosensors-12-00952-f005:**
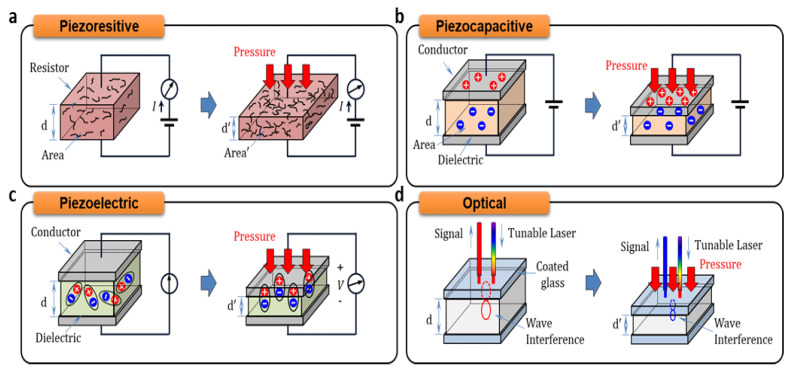
Schematic illustration of various mechanisms for biodegradable pressure sensors. (**a**) Piezoresistive type. (**b**) Piezocapacitive type. (**c**) Piezoelectric type. (**d**) Optical type.

**Figure 6 biosensors-12-00952-f006:**
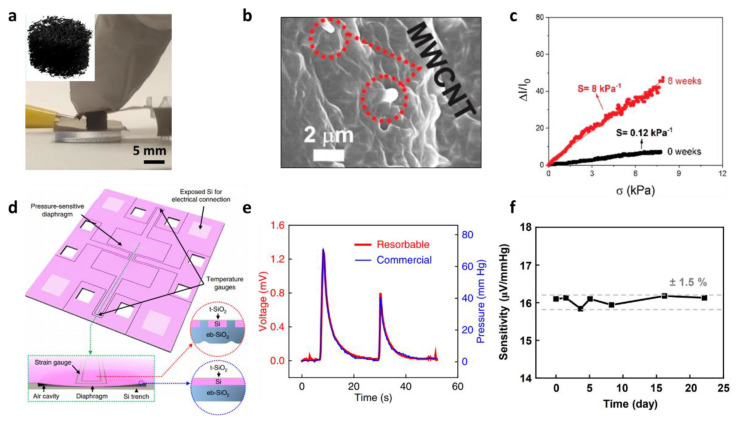
Biodegradable resistive pressure sensors. (**a**) Optical image of the piezoresistive foam sensor (inset: multi-walled carbon nanotube (MWCNT)-Poly (glycerol sebacate) (PGS). (**b**) Scanning electron microscope (SEM) image of the MWCNT-PGS foam. (**c**) Pressure-current response curve of the MWCNT-PGS foam device black), and its transition in the PBS solution (red) [127]. (**d**) Schematic illustration of bioresorbable pressure sensors protected with a thermally grown silicon dioxide (t-SiO_2_) nano-layer. Inset: cross-section across the diaphragm revealing the air cavity and silicon (Si) trench located underneath. The cross-sections across the strain gauge (red inset) and non-strain gauge (blue) regions showing the tri-layer composition. (**e**) Responses of the fabricated device (red) and commercial sensor (blue) to time-varying pressures in in vitro evaluations. (**f**) Comprehensive results from continuous in vitro operation over a period of 22 days (variations in pressure sensitivity within ±1.5%) [30].

**Figure 7 biosensors-12-00952-f007:**
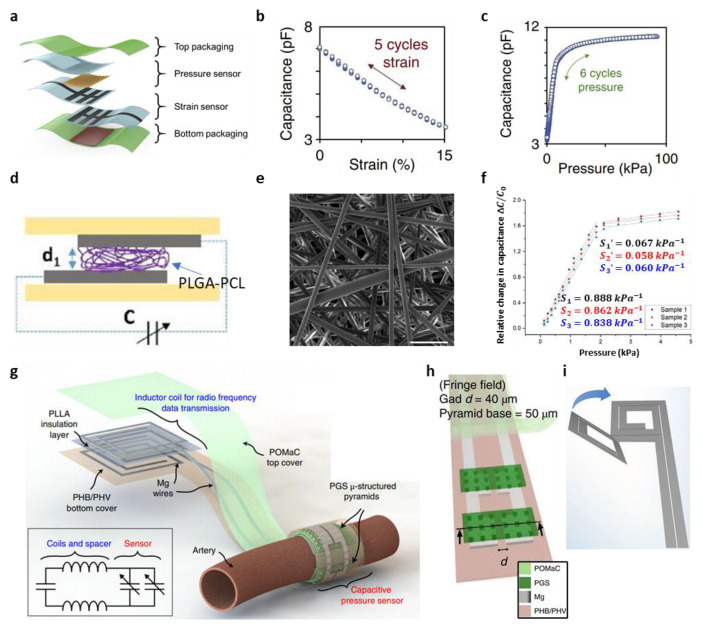
Biodegradable capacitive pressure sensors. (**a**) Schematic illustration of an integrated strain and pressure sensor. (**b**) Strain and capacitance-response in loading-unloading cycle (5-cycle). (**c**) Cyclic test of pressure and capacitance-response (6-cycle) [131]. (**d**) Schematic illustration of a biodegradable pressure sensor based on nano-fibrous dielectric. (**e**) SEM image of the polylactic-co-glycolic acid (PLGA)-polycaprolactone (PCL) layer (Scale bar, 10 μm). (**f**) Pressure and capacitance-response curve of the device [133]. (**g**) Schematic illustration of a biodegradable, flexible and passive arterial-pulse sensor. Inset: equivalent circuit of the device (left). (**h**) Flexible design of the pressure-sensitive region of the device. (**i**) Optimal design of the inductor coil (asymmetrical; the top coil turns clockwise, whereas the bottom coil turns anticlockwise) [134].

**Figure 8 biosensors-12-00952-f008:**
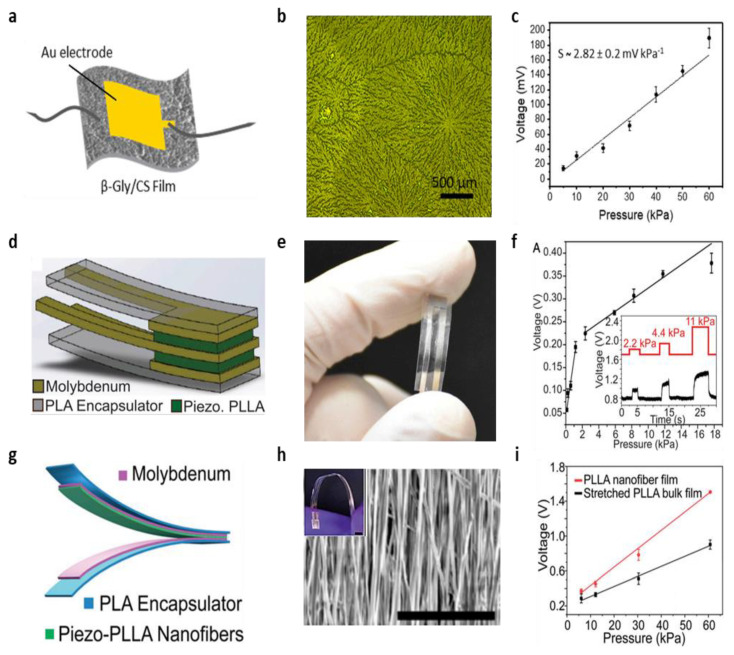
Biodegradable piezoelectric pressure sensors. (**a**) Design of β-glycine-chitosan (Gly/CS)-based flexible biodegradable piezoelectric pressure sensor. (**b**) Optical microscope image of the β-Gly/CS film. (**c**) Piezoelectric sensitivity of the sensor as a function of applied pressure [136]. (**d**) Schematic illustration of a biodegradable piezoelectric PLLA pressure sensor. (**e**) Optical image of the fabricated device (5 mm × 5 mm and 200 μm thick). (**f**) Pressure-response curve generated by a poly (L-lactic acid) (PLLA) sensor. Inset shows output voltage signals from different input pressure [122]. (**g**) Schematic illustration of a biodegradable nanofiber-based piezoelectric pressure sensor. (**h**) SEM image of piezo-PLLA nanofibers (Scale bar, 40 μm) (inset: the flexible fabricated device). (**i**) Comparison of calibration curves for a biodegradable sensor using stretched, bulk piezo-PLLA film (black) and a PLLA nanofiber film (red) [123].

**Figure 9 biosensors-12-00952-f009:**
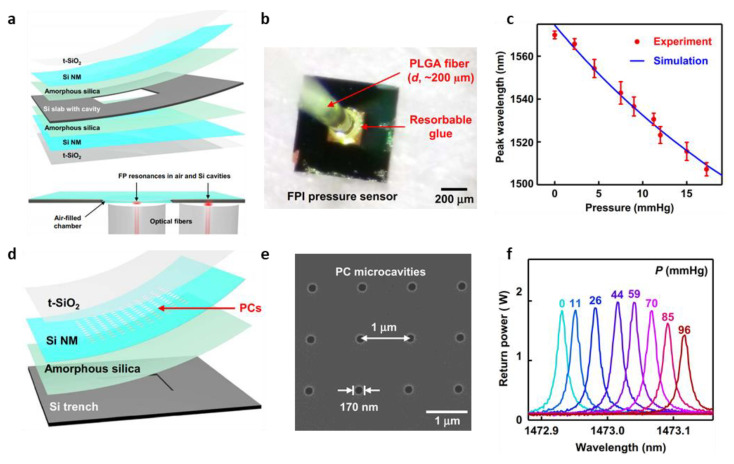
Biodegradable optical pressure sensors. (**a**) Schematic illustration of a biodegradable Fabry–Perot interferometers (FBI) pressure sensor (top). A cross-sectional view of the sensor integrated with two optical fibers (bottom). (**b**) Optical image of a PLGA fiber and the device. (**c**) Pressure-peak wavelength-response curve for the device (red) compared with simulation results (blue). (**d**) Schematic illustration of a biodegradable photonic crystal structure (PCs) pressure sensor. (**e**) SEM image of microcavities of PCs. (**f**) Pressure–wavelength response of the device (red) compared with simulation data (blue) [143].

**Figure 10 biosensors-12-00952-f010:**
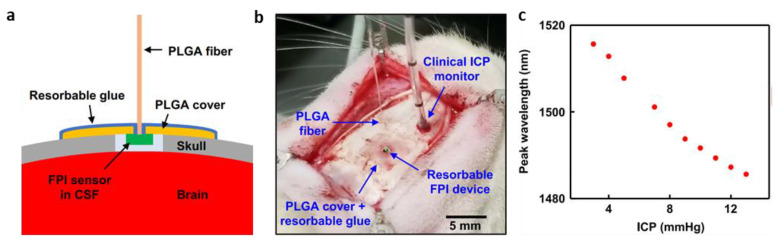
Application of biodegradable pressure sensor for the intracranial pressure (ICP) monitoring. (**a**) Cross-sectional schematic illustration of the device setup for animal-test. (**b**) Photograph of a bioresorbable FPI sensor implanted in the intracranial space of a rat for monitoring ICP. (**c**) Pressure calibration curve obtained from a bioresorbable FPI pressure sensor [143].

**Figure 11 biosensors-12-00952-f011:**
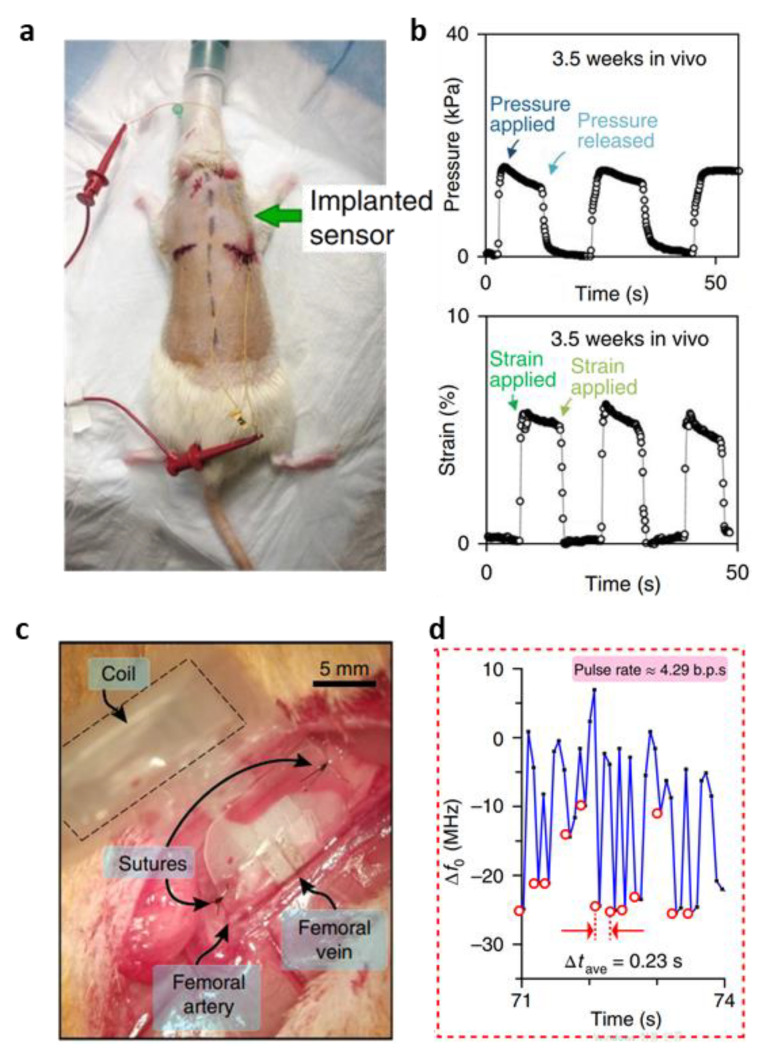
Other biomedical applications of the biodegradable pressure sensor. (**a**) Optical image of animal-test (rat model) implanted with the biodegradable strain and pressure sensor. (**b**) Measured pressure (top) and strain (bottom) signal recorded 3.5 weeks after sensor implantation [131]. (**c**) Optical image of an animal-test (rat model) implanted with a biodegradable and flexible arterial-pulse sensor. (**d**) $\Delta f_{0}$ measured on the femoral artery displaying an average pulse rate of 4.29 b.p.s. [134].

**Figure 12 biosensors-12-00952-f012:**
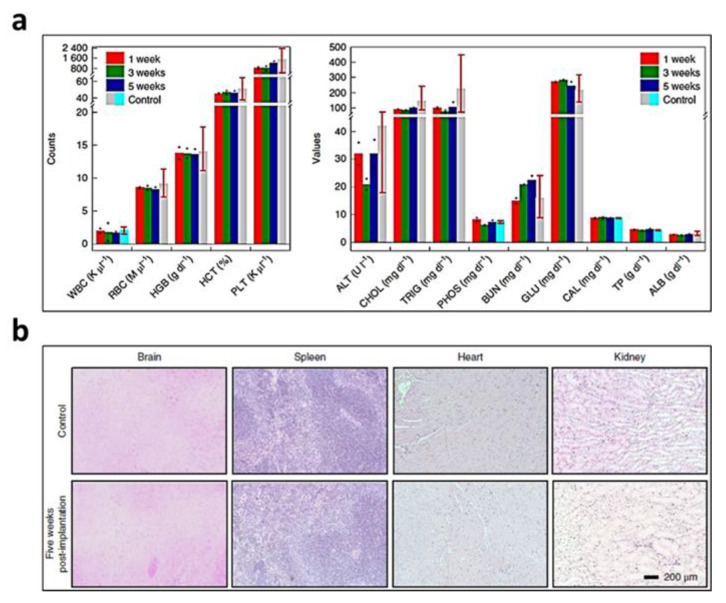
Studies of biocompatibility of biodegradable pressure sensors. (**a**) Results of the complete blood count and blood chemistry tests for the mice implanted with the biodegradable resistive pressure sensor. (**b**) Histopathologic evaluation of the biodegradable resistive pressure sensor in various tissue [30].

**Table 1 biosensors-12-00952-t001:** Summary of characteristics of biodegradable pressure sensors.

Ref.	Key Features	Mechanism	Key Structure	Target	Sensitivity (Pressure Range)	Functional Lifetime
[127]	Ultra sensitivity (8 kPa^−1^)	Resistive	MWCNT-PGS, Nanocomposite, Morphology and Porosity	E-skin	8 kPa^−1^ (0 < *p* < 8.5 kPa) (In a PBS solution)	N/A
[39]	Multi Si sensors	Resistive	Si-Nanomembrane	Intracranial pressure; Intra-abdominal cavity; Leg cavity	622.55 Ω/kPa (0 < *p* < 10.67 kPa)	3 days (In vivo)
[30]	Long-Functional lifetime (25 days)	Resistive	Si-Nanomembrane, t-SiO_2_ layer	Intracranial pressure	0.98 Ω/kPa (0 < *p* < 5.33 kPa) (In vivo)	Fine on day 18; 4 mmHg drifts on day 25 (In vivo)
[40]	Fast dissolution of Si MM barrier; Wax edge barrier; Warning indicators	Resistive	Si-Nanomembrane	Intracranial pressure	118.51 Ω/kPa (0 < *p* < 13.87 kPa) (In vitro)	3 weeks (In vivo)
[64]	Wireless; RF	Capacitive	Zn/Fe bilayer conductor	Implantation application	39 kHz/kPa (0 < *p* < 20 kPa) (In saline)	86 h (In saline)
[47]	Microstructured pyramid; Sensing array	Capacitive	PGS pyramid structure of dielectric	Cardiovascular pressure	0.76 ± 0.14 kPa^−1^ (0 < *p* < 2 kPa)	N/A
					0.11 ± 0.07 kPa^−1^ (2 < *p* < 10 kPa)	
[134]	Microstructured pyramid; Contact/Non-contact mode; Wireless	Capacitive	PGS pyramid structure of dielectric	Arterial pulse	N/A	N/A
[133]	Nanofibers	Capacitive	PLGA-PCL nanofibers	E-skin	0.863 ± 0.025 kPa^−1^ (0 < *p* < 1.86 kPa)	N/A
					0.062 ± 0.005 kPa^−1^ (1.86 < *p* < 4.6 kPa)	
[37]	Wireless	Capacitive	Mg Coil & electrode	Orthopedic application	−45.00 ± 3.75 kHz/kPa (0 < *p* < 26.66 kPa) (In vitro)	8 h (In vitro)
[135]	long-Functional lifetime of Wireless sensor (4 days)	Capacitive	Mg Coil	Intracranial pressure	1500.12 kHz/kPa (0 < *p* < 4 kPa) (In vitro)	4 days (In vivo)
[131]	Microstructured pyramid; Strain and pressure sensor	Capacitive	PGS pyramid structure of dielectric	Orthopedic application	0.7 ± 0.4 kPa^−1^ (0 < *p* < 1 kPa)	3.5 weeks (In vivo)
					0.13 ± 0.03 kPa^−1^ (5 < *p* < 10 kPa)	
[122]	Maximal piezoelectric output of PLLA; bilayer	Piezoelectric	treated PLLA	Intra-abdominal cavity	~0.12 V/kPa (0 < *p* < 2 kPa) ~0.013 V/kPa (2 < *p* < 18 kPa)	4 days (In PBS)
[123]	Bilayer	Piezoelectric	PLLA nanofiber film	Intra-abdominal cavity	N/A	N/A
[136]	Organic piezoelectric material	Piezoelectric	β-Gly/CS film	E-skin	2.82 ± 0.2 mV/kPa (5 < *p* < 60 kPa)	N/A
[143]	Compatibility with MRI Wireless	Optical	Si-Nanomembrane; PC-microcavities; t-SiO_2_ layer	Intracranial pressure	23.25 mm/kPa (0.40 < *p* < 1.73 kPa) (In vitro)	7 days (In vitro)

## Data Availability

Not applicable.

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
