# Peer review of "Micro-/Nano-Structured Biodegradable Pressure Sensors for Biomedical Applications"

_biosensors, 2022, doi:10.3390/bios12110952_

Round 1
Reviewer 1 Report
The authors have reviewed a compilation of the most recent biodegradable sensors for medical applications. Their selection contains the most up-to-date technology on biocompatible materials that degrade in the environment of the human body and that have been used to measure pressure in medical applications. Their study includes material discussion and mechanisms of degradation, medical applications and biocompatibility studies required to validate such devices.
The topic of the article is of interest and the discussions offered are structured and scientifically appropriate. The specificity of the topic, i.e., pressure sensors for biomedical devices, may be slightly forced, as there are already some reviews of biodegradable sensors for medical applications. Nevertheless, the authors have proven that the topic has enough developments to provide a complete state of the art review.
Perhaps the weakest section of the manuscript is the biomedical applications one, as describing only the medical application whereby reading pressures would be of significance seems a bit incomplete. The designs of the sensors are usually heavily influenced by the application addressed, and therefore separating design and application seem to leave the examples cited a bit isolated and maybe without proper context.

Author Response
The authors would like to appreciate sincerely for all of your editorial efforts and kind reply on our manuscript. We have received the letter containing valuable reviewers` comments and their keen reviews were very impressive and helpful to the authors. We understand this editorial process makes our paper healthier. We have done our best to answer all the reviewers` questions, and revised our manuscript and supporting information, accordingly. All amended sentences were underlined here and marked in the revised manuscript.

Reviewer 2 Report
This review article reviews the recent progress in micro-/nano-structured biodegradable pressure sensor devices. The authors discuss the material selection and sensing mechanisms for the preparation of biodegradable pressure sensors, and present recent advances in biodegradable pressure sensors and their applications in biomedical fields. There are some major comments that need to be addressed before the article is published.
1. Some of the materials presented in section 2.2 can hardly be considered as biodegradable. It is difficult to meet the conditions for their degradation in living organisms. For example: a) When exposed to sufficient heat (~55℃), the device broke down rapidly as the protective wax coatings melted and released the encapsuled acid microdroplets. b) The PAMS layer releases volatile monomers (α-methylstyrene) when it is heated up to ~250–300℃.
2. In section 3, the authors simply list the papers and work, lacking the corresponding summary work.
3. In section 3.2.3, there are only three references, which is too few. Three references can hardly summarize the progress in this direction.
4. The title of the review is Micro-/Nano-structured Biodegradable Pressure Sensors for Biomedical Applications, but there are few summaries of the specific structures and processes of micro-nano-structures.
5. The review lacks summary and outlook on the research direction of biodegradable pressure sensors.
Author Response

(The authors gave the same response as above.)

Reviewer 3 Report
Please see the attached file

Author Response

(The authors gave the same response as above.)

Round 2
Reviewer 2 Report
I have no questions with this manuscript.